# Everyone Can Implement Eduball in Physical Education to Develop Cognitive and Motor Skills in Primary School Students

**DOI:** 10.3390/ijerph19031275

**Published:** 2022-01-24

**Authors:** Sara Wawrzyniak, Marcin Korbecki, Ireneusz Cichy, Agnieszka Kruszwicka, Tomasz Przybyla, Michal Klichowski, Andrzej Rokita

**Affiliations:** 1Department of Team Sports Games, Wroclaw University of Health and Sport Sciences, Mickiewicza 58, 51-684 Wroclaw, Poland; marcin.korbecki@hat-bud.pl (M.K.); ireneusz.cichy@awf.wroc.pl (I.C.); andrzej.rokita@awf.wroc.pl (A.R.); 2Learning Laboratory, Faculty of Educational Studies, Adam Mickiewicz University, Szamarzewskiego 89, 60-568 Poznan, Poland; a.kruszwicka@gmail.com (A.K.); tomekprzybyla@gmail.com (T.P.)

**Keywords:** educational balls, gross motor, learning, locomotor skills, mathematical skills, object control, primary education, reading, writing

## Abstract

Studies suggest that incorporating core academic subjects into physical education (PE) stimulates the development of both motor and cognitive skills in primary school students. For example, several experiments show that children’s participation in Eduball, i.e., a method that uses educational balls with printed letters, numbers, and other signs, improves their physical fitness while simultaneously developing their mathematical and language skills. However, the question of who should conduct such classes to make them most effective (regular classroom teachers, physical education teachers, or maybe both in cooperation?) remains unanswered. Here, we replicated a previous Eduball experiment, but now, instead of one experimental group, there were three. In the first, Eduball-classes were conducted by the classroom teacher, in the second, by the physical education teacher, and in the third, collaboratively. After one year intervention, all experimental groups significantly improved both their cognitive (mathematical, reading, and writing) and gross motor (locomotor and object control) skills, and these effects were larger than in the control group participating in traditional PE. Importantly, there were no differences in progression between the Eduball-groups. Thus, our study demonstrates that methods linking PE with cognitive tasks can be effectively used by both PE specialists and general classroom teachers.

## 1. Introduction

A vast amount of literature (e.g., concerning the idea of embodied cognition or multi-sensory integration) assumes strong interactions between action and cognition and postulates that they should not be developed, in education, independently of each other [1,2]. From the perspective of psychophysiology and neuroscience, this is explained, for example, by the fact that the neurophysiological mechanisms of movement are at the core of cognition, and the areas of the brain related to planning movements are closely linked to those associated with cognitive processes [3,4,5]. Therefore, in primary school, it is becoming more and more common to integrate physical education (PE) with core academic subjects. Numerous studies [6,7,8,9,10,11] confirm that such an approach stimulates not only physical and health literacy or movement proficiency but also cognitive and social skills and, consequently, influences children’s academic achievements (see also [12]). For example, in several experiments [13,14,15,16,17,18,19], it has been found that participation in PE with Eduball, i.e., a method that uses educational balls with printed letters, numbers, and other signs, improves children’s gross motor and graphomotor skills, while simultaneously developing their mathematical competencies, as well as reading and writing abilities. Therefore, the Eduball method is used in hundreds of schools in Europe, the United States [15,16], and Asia [17]. However, a key question is who should conduct such classes (Eduball-based or based on other methods integrating PE with cognitive tasks) to make them most effective. In many countries, PE in primary school is taught mostly by a regular classroom teacher (CT) who is formally qualified to teach all, or almost all, subjects in the curriculum, including PE [20]. Therefore, it seems that CT should be able to cope well with methods combining PE with core academic subjects. Nonetheless, PE may also be led by physical education teachers (PET), i.e., by a PE specialist. However, such a teacher is not usually qualified to teach core academic subjects. Perhaps then, in the case of linking PE with cognitive tasks, PET should cooperate with CT? Although there is ongoing debate regarding which of these three options may be most effective, no definitive (evidence-based) answer has been provided [18,21]. Thus, there is an unmet need for research on this issue.

Unfortunately, it is difficult to propose any directional hypothesis for such studies, i.e., indicating the superiority of one of these three approaches. Firstly, some findings [22] suggest that CTs are best placed to teach the child-centered, integrated curriculum in primary schools. However, other outcomes [23,24,25,26,27] demonstrate that many CTs are not confident and/or feel poorly prepared to teach PE programs. Secondly, further reports [23,28,29] indicate that PETs are more effective than CTs in teaching PE, i.e., specialist-taught PE supports children’s cognitive-motor development better than CT-taught PE. At the same time, other studies [11,30,31,32] show that PE may be best implemented when PET and CT work together, and only in this way, through supporting one another, will they have a real positive effect on students’ physical and intellectual performance. Nonetheless, there are numerous obstacles (for example, institutional or financial) to implementing such a collaborative approach in primary schools [22,25,26,27]. Finally, in our previous Eduball-experiments, whether PE classes were led by CT [15,18] or by PET [16,17], children from the experimental group significantly improved their academic and/or motor skills more than students from the control group. Nevertheless, the effects of these approaches have never been compared in one experiment.

Therefore, we postulate that integrating PE with core academic subjects can be effectively used by both CTs and PETs as well as in collaboration. To test this hypothesis, we replicated our earlier one-year Eduball experiment, but now, instead of one experimental group, there were three Eduball-groups: (1) conducted by CT, (2) conducted by PET, and (3) conducted together by CT and PET. We measured both motor and cognitive effects. Our results clearly show that, in the case of a proven method of integration of PE with cognitive tasks such as Eduball, each type of educator is skilled enough to implement it safely and effectively in PE lessons. Hence, our conclusion is that “everyone can”.

## 2. Materials and Methods

### 2.1. Participants

Seventy-three Polish students from four first-grade classes (33 girls, age: 6–7, mean = 6.32, *SD* = 0.47) participated in the experiment. All classes attended the same state school located in a large city in Poland. Classes were randomly assigned to one control (C) and three experimental groups (E1, E2, and E3). Three students did not participate in the post-test (one from E2, and two from E3). Therefore, the final research sample consisted of 70 students. The control group included 21 students (11 girls, mean age = 6.38, *SD* = 0.50), while E1 comprised 18 students (7 girls, mean age = 6.11, *SD* = 0.32), E2—16 students (5 girls, mean age = 6.06, *SD* = 0.25), and E3—15 students (8 girls, mean age = 6.80, *SD* = 0.41). To confirm that, at the beginning of our experiment, there were no significant differences between the groups in terms of their cognitive and motor skills (i.e., mathematical, reading, writing, locomotor, and object control skills), pre-test values were compared with a one-way ANOVA with *group type* as a factor. There was only a main effect for writing skills (*p* < 0.01), such that E3 scored higher than E2 (*Bonferroni-p* < 0.01, see Figure 3c and Table 1). No other main effects were found (all the remaining *p* > 0.13). In other words, there were no more differences in any of the categories between the experimental groups (all the remaining *Bonferroni-p* > 0.37), and—most importantly—the control group did not differ from any of the experimental group (all *Bonferroni-p* > 0.13).

### 2.2. Experimental Factor

The experimental factor was the Eduball method. The main didactic aid used in this method was a set of 100 balls in a size adapted to students aged 6–8 years. Further details on the Eduball set are presented in Figure 1.

As in the previous Eduball-experiments [15,16,17,18], various games from the Eduball-set [13,14] (see also supplementary materials from our recent Eduball-papers [15,16]) were incorporated into the PE classes’ scenarios. All games were adapted by the teachers conducting the classes (with the help of our team). The adaptation was aimed at integrating currently taught educational content in mathematics and Polish as native language lessons with Eduball exercises, according to the thematic cycle and the theme of the day (in compliance with the curricula). Therefore, when students practiced in lessons, for example, spelling (as part of the language lessons), the Eduball-games included activities related to orthographic rules. Furthermore, the core curriculum was respected while planning games and scenarios. Note that the process of customizing games to the program never changed Eduball’s idea, and each game reflected the basic principles of the method. Thus, all activities that made it possible to exercise motor and cognitive skills were played as a team. In other words, as part of each Eduball-game, students not only practiced physical skills (e.g., running, jumping, throwing, catching, passing and catching the ball, and many other physical abilities) but also learned a selected part of knowledge related to the native language and mathematics (e.g., letters, spelling, reading, combining letters into syllables, counting, arithmetic operations, punctuation marks, and language rules). As such, during each single lesson, several adapted games from the Eduball-set were carried out, and all experimental groups followed similar scenarios. For example, the game “Letter Tag with the Letter M” [13], which aimed to improve the ability to create words beginning with the letter “M” and shape selected motor and movement skills, was conducted with all groups in the first week of October.

### 2.3. Procedure

The procedures for this study were assessed and approved by the local Ethics Committee for Research Involving Human Subjects (Resolution #37/2016 of the Senate Committee on Ethics of Scientific Research at the Wroclaw University of Health and Sport Sciences on 16 October 2016). The experiment was conducted in accordance with the principles of the Helsinki Declaration.

The experiment lasted 10 months and was performed during the whole school year (which, in Poland, begins in September and ends in June) in natural conditions (at school) using parallel groups. The experimental groups (E1, E2, and E3) and the control group (C) followed the same integrated curriculum: “Our Elementary: Autumn, Winter, Spring, Summer”, which utilizes the method of weekly thematic projects [33,34]. In all groups, there were three 45-min PE classes per week. In the experimental groups, two of them were enhanced with Eduball (as such, 68 units of Eduball were taught). As in our previous Eduball-experiments [15,16,17,18], in the control group, all PE classes (102 units) were held without Eduball and followed the standard PE program. Thus, the teacher of C conducted the PE program in accordance with the aims and objectives of the school’s program for developing physical fitness and health education. In C and E1, PE classes were led by CT, while in E2, they were taught by PET, and in E3, by both collaboratively. This cooperation involved the CT conducting the class but in consultation with PET, who had a strong advisory role. PET suggested, for example, changes in the intensity of activity and the sequencing of tasks. In other words, in E3, CT and PET prepared classes together, but CT conducted them.

Our study included two measurement periods: a pre-test at the beginning of the school year (third and fourth week of September) and a post-test at the end of the school year (first and second week of June). During both periods, fundamental motor skills were diagnosed using the Test of Gross Motor Development (Second Edition) [35] and school achievement was assessed using the Test of School Start Skills [36]. Pre-test and post-test were carried out in the same order: we tested fundamental motor skills first, and then, we diagnosed school achievement. The assessment of each test was always done by experimenters. The scheme of our experimental workflow is depicted in Figure 2.

#### 2.3.1. Test of Gross Motor Development—Second Edition (TGMD-2)

The Test of Gross Motor Development—Second Edition (TGMD-2) was used to determine fundamental movement skills. The TGMD-2 consists of two subtests: locomotor skills (run, gallop, hop, leap, jump, and slide) and object control (strike, dribble, catch, kick, throw, and roll) [35].

The TGMD-2 testing was conducted during a school PE class at the sports hall by four experimenters (one supervisor and three doctoral students of the Wroclaw University of Health and Sport Sciences). First, one experimenter demonstrated the proper execution of the locomotor and object control skills, and then, students re-demonstrated these actions in the same order (locomotor skills, object control skills). Every participant had to complete one practice and then two formal trials. All experimenters observed and scored each re-demonstration for each trial on the spot, based on three to five performance criteria (e.g., for gallop: arms bent and lifted to waist level at take-off; a step forward with the lead foot followed by a step with the trailing foot to a position adjacent to or behind the lead foot; a brief period where both feet are off the floor; maintaining a rhythmic pattern for four consecutive gallops; for more examples see [37]). If a student demonstrated the correct performance criteria, a score of 1 for each criterion was given; otherwise, he/she received a score of 0. The highest total raw score for the two subtests was 48 points. The higher the total score, the better the performance.

#### 2.3.2. Test of Skills at School Start (TUNSS)

The Test of Skills at School Start (TUNSS) was used to assess participants’ academic performance. This test is commonly used in Poland. It consists of three measurement scales: mathematical skills scale (numbers, measurement, space and shape, and relationships and dependencies), writing skills scale (visual-motor skills, visual-spatial skills, auditory-linguistic skills, and calligraphy), and reading skills scale (auditory-linguistic, auditory-visual, and reading skills) [36].

To ensure adequate focus, good lighting, and to delineate background noise, TUNSS was conducted in the school teacher’s office (near students’ classrooms). During the test, only the participant and experimenter were present. The heights of the chair and the table were adjusted to the age of the participants. On the latter, there was a 10.1-inch Samsung Galaxy Tab 2 P5110 (16GB, Samsung Group, Seoul, Korea), as well as pencils and worksheets. TUNSS began with the introductory part, which includes establishing contact with a given student, introducing the aim of the test, and completing the tutorial task. During this time, the participant was familiarized with how to use the tablet and the TUNNS-application. Simultaneously, TUNSS adapted the level of difficulty and the number of tasks to the student’s knowledge and skills. After this phase, each student solved not less than five and not more than ten tasks for every scale. All tasks were performed on the tablet with a finger, but for the writing scale, the special cards were also used (here, with a pencil). Finally, a score for each scale was calculated and presented in values similar to those for the IQ test.

### 2.4. Data Analysis

The main dependent variables were represented by mathematical, reading, and writing skills, as well as locomotor and object control skills. These variables were expressed as mean scores and calculated separately for the control and experimental groups, as well as for pre- and post-test. Data analysis was as described in our previous study [17]. In short: first, a paired samples *t*-test was applied to compare the changes in the score (pre-test vs. post-test) within the control and experimental groups. Here, an effect size was calculated using Cohen’s *d* and was interpreted as: <0.2—very small or no effect, 0.2–0.5—small, 0.5–0.8—medium, 0.8–1.2—large, 1.2–2.0—very large, and >2.0—huge [38]. Second, two analyses of covariance (ANCOVAs) were run to determine the significant difference between groups after the experiment (control group vs. all experimental groups taken together, and E1 vs. E2 vs. E3). The students’ post-test scores were set as the dependent variable and the pre-test scores as the covariate. Finally, if necessary, Bonferroni-corrected post-hoc tests were calculated. The adopted level of significance was α = 0.05. IBM^®^ SPSS Statistics^®^ for Mac Version 27.0 (IBM Corp., Armonk, NY, USA) was used for all statistical analyses. All the anonymized raw data acquired in this project are publicly available in the Open Science Framework at https://osf.io/fhqpj (accessed on 20 December 2021).

## 3. Results

As shown in Table 1, after one school year, all groups improved the level of their mathematical, reading, and writing skills. When we analyzed the total score, it was the same for locomotor and object control skills. However, analysis of the subscales showed that there were variations in the number and type of improved categories between groups. The control group improved in jumping, dribbling, and throwing, while E1 improved in hopping, jumping, sliding, dribbling, and catching; E2 in jumping, dribbling, kicking, and rolling a ball; and E3 in galloping, leaping, and dribbling.

When analyzing the effect sizes (*d* in Table 1), it was observed that improvements were larger in the Eduball-groups as compared to the no-Eduball one. In the control group, effect sizes were large for the cognitive categories (mathematical, reading, and writing skills) and medium for the gross motor ones (locomotor and object control skills). In E1, E2, and E3, effect sizes for the cognitive categories were huge or very large, and very large or large for locomotor skills. In E2 and E3, effect sizes were also large for object control skills (in E1, effect size was medium, as in C). These results are shown in Figure 3.

The first one-way ANCOVA was conducted to test the effectiveness of the Eduball program in terms of its influence on the students’ cognitive and gross motor skills. Therefore, we combined all the experimental groups into one (Eduball) group and compared their aggregate results with the results of the control (no-Eduball) group (see Figure 4). As depicted in Table 2, there were significant differences in reading and writing skills, as well as in total locomotor skills (and in galloping and hopping in subscale) between the groups, with the Eduball group making more progress relative to the no-Eduball one. In the case of mathematical and total object control skills (and only in dribbling), we found similar effects. However, these were just trends. These results are expressed as difference scores, relative to the pre-test baseline (“0”), in Figure 5.

The second one-way ANCOVA was performed to determine whether the Eduball method could be used effectively by both CTs and PETs, as well as in collaboration between the two types of educators. Thus, we compared the results of E1, E2, and E3. As predicted, there were no significant differences in mathematical, reading, and writing skills, as well as in total locomotor and total object control skills. Only for one subscale of locomotor skills and one of object control skills we found differences; namely for sliding, E1 made progress compared to E2 (*Bonferroni-p* < 0.01) and E3 (*Bonferroni-p* < 0.001), and for catching, E1 > E2 (*Bonferroni-p* = 0.02). The results of these comparisons are shown in Table 3 and expressed as difference scores, relative to the pre-test baseline (“0”), in Figure 6.

## 4. Discussion

Our study shows that one-year participation in PE classes with Eduball, i.e., a method that uses educational balls with printed signs to incorporate core academic subjects into PE, stimulates the development of both gross motor and cognitive skills in primary school students more than participating in traditional PE. We observed this effect, mainly, for locomotor skills, as well as reading and writing skills, as has already been demonstrated by previous Eduball-experiments. Most importantly, similar motor and cognitive progressions were found in all experimental groups, which confirms our hypothesis that methods linking PE with cognitive tasks can be effectively used by both CT and PET and, moreover, collaboratively. Therefore, our results not only support the theoretical framework of holistic education but they can serve as a guideline for the organization of a child’s education.

### 4.1. Eduball and Motor Skills

The results of a paired *t*-test comparison of pre- and post-test scores for total locomotor and object control skills are consistent with the observations from earlier Eduball [15,16,17,18,39] and non-Eduball experiments [6,8,10,11,40], which showed that incorporating cognitive tasks into PE does not weaken the motor effects of PE but stimulates physical development. Nevertheless, in the current project, the differences between the control and experimental groups, revealed by ANCOVA, were not as great as in the previous Eduball-study in which the same test of gross motor development (TGMD-2) was used [17]. In that investigation, we observed differences in both total locomotor skills and total object control skills. In this study, there was only a trend for the latter. This is a surprise, because the literature review by Morgan et al. [41] showed that, generally, innovations during PE stimulate the development of object control skills the most (however, another review [42] indicated that, overall, a small number of studies measured object control skills). Likewise, there were also discrepancies for subcategories of TGMD-2. In the previous experiment, we found significant differences between the no-Eduball and Eduball groups for running, galloping, striking, kicking, and rolling. In this study, such differences were detected only for galloping, hopping and dribbling. This is probably due to the fact that very young students (6–7 years old) from the first grade of primary school participated in this experiment. Previously, it was either the second or third grade, or if the first grade, only with seven-year-olds. At the time of this study, in Poland, children aged six and seven started their education at school together, in age-balanced classes [43]. From earlier studies using TGMD-2 [44], we know that students who start school at the age of six make less progress after the first year of education in gross motor skills, particularly in object control skills (e.g., in rolling), than those who start at the age of seven. As 68% of our sample was six years old, the scores they obtained may have affected the group results. Our sample size is, however, too small to be divided into subgroups, so we cannot establish whether or not this was the case. Future research should take this issue into account with a greater sample size. Note that, now, this policy (concerning six-year-olds starting school) has been abandoned in Poland, and six-year-olds learn in pre-schools, while only seven-year-olds enter the first year of primary school [45].

### 4.2. Eduball and Cognitive Skills

Our study demonstrates that cognitive-enriched PE lessons, such as those conducted using the Eduball method, have a positive effect on children’s academic performance. More precisely, after the experiment, we observed higher progress in the Eduball-groups, both in mathematical and language skills, compared to the control one. These findings are in line with our previous research [15,16,18], in which we investigated the influence of Eduball on academic performance in primary school students, as well as with other (non-Eduball) studies (e.g., [38,46]) involving cognitive tasks in PE. The above-mentioned observations can be explained by the strong language-mathematics and motor developmental relationships, i.e., similar mechanisms of these functions at the neuronal level [47,48,49]. It is worth noting that children learn to count by using their fingers [50]. Likewise, language is acquired (to some extent) through movement; e.g., first words are preceded by gestures, and the development of speech is tightly related to gesticulating [51]. As such, specific motor training (e.g., hands training with forced use of the non-dominant hand [52]) increases young children’s communication and numerical skills [53,54,55,56]. Our results, therefore, not only once again confirm the practice of combining PE with cognitive tasks but also support the concept of embodied mathematical and linguistic cognition [57,58,59].

### 4.3. Eduball and Teacher Types

Our outcomes clearly show that the Eduball method brings the expected physical-cognitive results in primary school no matter who is teaching the Eduball-classes. In other words, the present project confirms the hypothesis that proven approaches linking PE with core academic subjects can be effectively used by both CTs and PETs, as well as collaboratively. These findings are in line with past Eduball-studies [15,16,17,18,19] in which we observed motor and cognitive increases in students’ learning, both when CT and PET led PE. However, they contradict some reports on diverse methods [22,23,28,29,30,31,32], which have suggested one type of teacher is superior in the effective delivery of PE. Thus, it seems it is not the type of educator that is the issue, but the type of intervention/method used. Battaglia et al. [60] emphasize that, because of the complexity of children’s dexterities, it is very hard to find suitable PE methods that can stimulate students’ development of cognitive and motor skills. Another difficulty is the fact that such methods should provide teachers with clear guidance, and teaching aids dedicated to them must be durable and easy to use [61,62]. Furthermore, there is a need to break down the barriers faced in teaching PE, such as personal school experience in PE (most often separated from cognitive effort), uncertainty about how to integrate a variety of academic contents into movement (most frequently not taught in pre-service teacher training), and level of departmental assistance (usually a complete lack of support in the integration of PE with cognitive activity) [26,63,64,65]. Our investigation suggests that Eduball may have clear potential in this regard, but further research is still needed on this issue. For example, research should be conducted with older students and using other tests. In addition, graphomotor skills and various fine motor abilities should be measured. Likewise, it would be of interest to test one extra control group using another non-traditional (mixing PE with intellectual activity) method.

## 5. Conclusions

Although numerous studies have shown that PE classes with cognitive elements, such as counting, reading, or creating words, have many benefits for the physical and intellectual development of children, questions remained over who should conduct them. Therefore, we compared the effectiveness of the Eduball-method, which links PE with cognitive tasks, when it was taught by CT and by PET, as well as by both collaboratively. Our results show that such methods can stimulate the development of both gross motor and cognitive skills in primary school students, irrespective of which teacher incorporates them. Thus, this project not only confirms, once again, the validity of combining PE with cognitive activity at school but also indicates that every teacher should be able to effectively use methods such as Eduball for this purpose.

## Figures and Tables

**Figure 1 ijerph-19-01275-f001:**
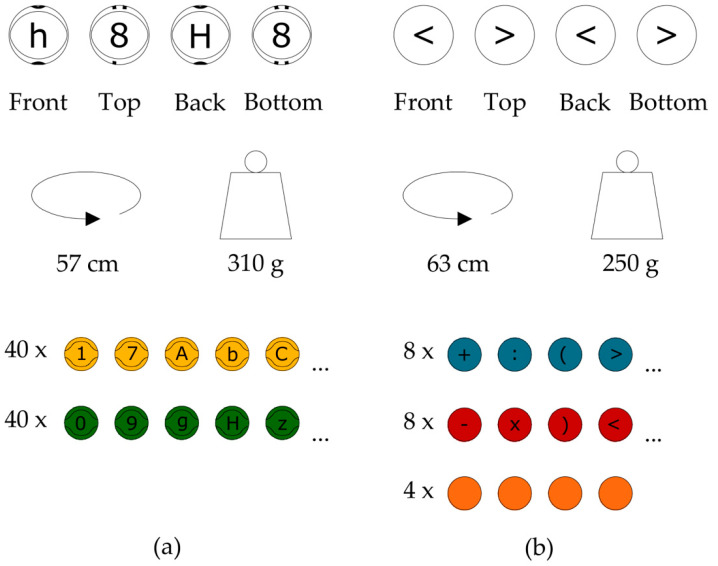
Eduball set. Balls used in the experiment were divided into two main subcategories. (**a**) The first type of balls (57 cm in circumference and 310 g of weight; close to size 3 basketballs) are in yellow and green colors (40 in each color) on which are printed black letters (on one side uppercase and on the opposite side lowercase) and numbers (from 0 to 9; the same digit on the top and bottom side); (**b**) the second type of balls (63 cm in circumference and 250 g of weight; close to size 4 volleyballs) are in blue and red colors (eight in each color) on the surfaces of which mathematical symbols are painted (representing the operations of addition, subtraction, multiplication, division, the symbols of greater than, less than, and parentheses), including the “at” sign (@), as well as four unprinted orange balls that can be used as universal blanks.

**Figure 2 ijerph-19-01275-f002:**
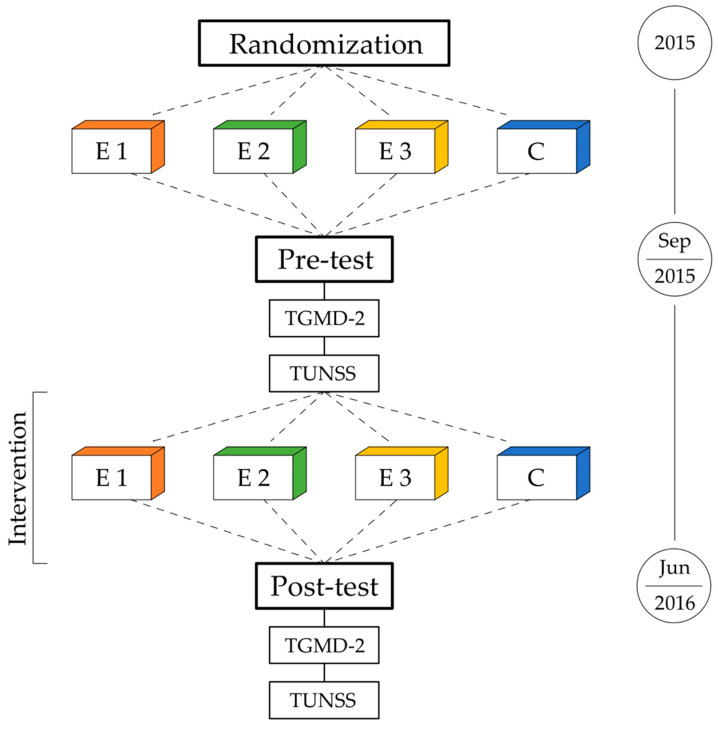
Schematic depicting the experimental workflow. Classes were randomly assigned into experimental (E1, E2, and E3) and control (C) groups. Pre-test was carried out as follows: (1) Test of Gross Motor Development (Second Edition) and (2) Test of School Start Skills. PE classes in C and E1 were conducted by a classroom teacher, while in E2 by a physical education teacher, and in E3 by both of them in cooperation. Post-test was carried out in the same order as pre-test.

**Figure 3 ijerph-19-01275-f003:**
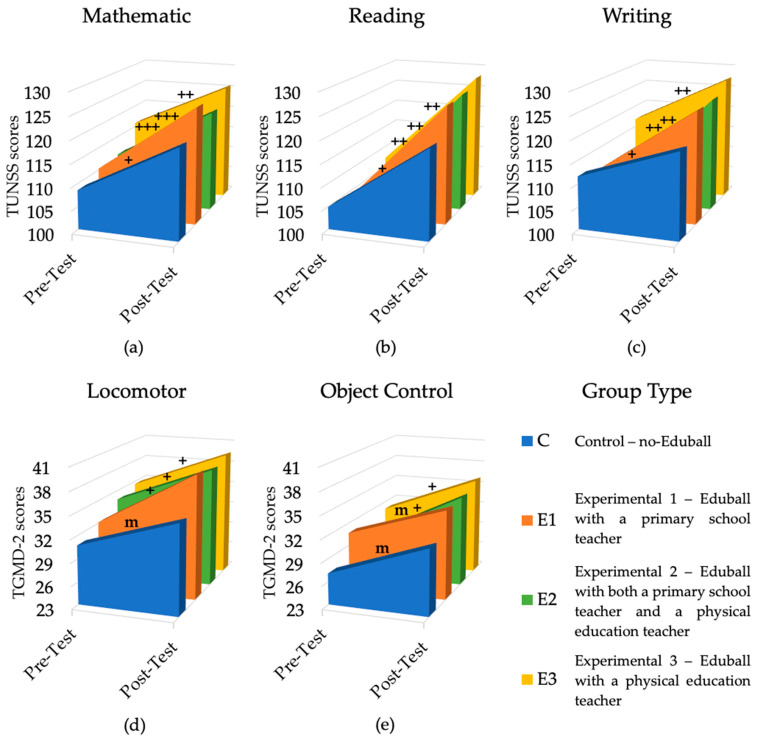
Results of the experiment (pre-test vs. post-test), in terms of cognitive and motor skills, divided into four groups (no-Eduball vs. Eduball 1–3). (**a**) The results of the four groups in terms of mathematical skills; (**b**) the results of the four groups in reading skills; (**c**) the results of the four groups in terms of writing skills; (**d**) the results of the four groups in terms of total locomotor skills; (**e**) the results of the four groups in terms of total object control skills. Plus (+) symbols reflect effect size: + large effect, ++ very large effect, +++ huge effect (m—medium effect).

**Figure 4 ijerph-19-01275-f004:**
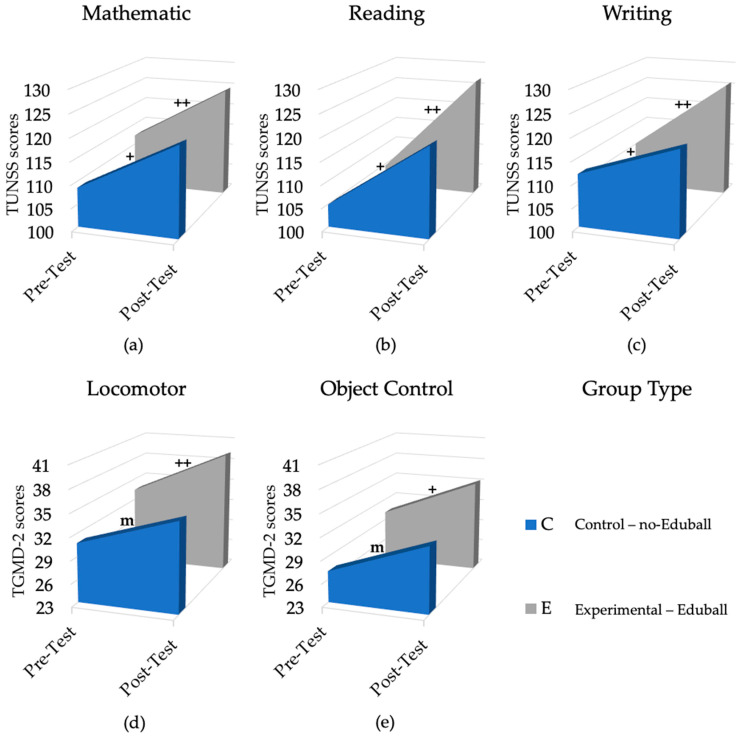
Results of the experiment (pre-test vs. post-test), in terms of cognitive and motor skills, divided into two groups (no-Eduball vs. Eduball). (**a**) The results of the two groups in terms of mathematical skills; (**b**) the results of the two groups in terms of reading skills; (**c**) the results of the two groups in terms of writing skills; (**d**) the results of the two groups in terms of total locomotor skills; (**e**) the results of the two groups in terms of total object control skills. Plus (+) symbols reflect effect size: + large effect, ++ very large effect (m—medium effect).

**Figure 5 ijerph-19-01275-f005:**
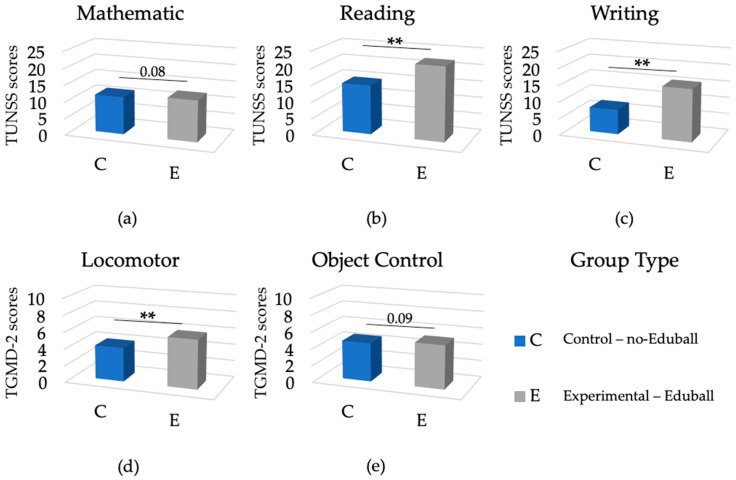
Results of the experiment, in terms of cognitive and motor skills, divided into two groups (no-Eduball vs. Eduball) and expressed as difference scores. (**a**) The results of the two groups in terms of mathematical skills; (**b**) the results of the two groups in terms of reading skills; (**c**) the results of the two groups in terms of writing skills; (**d**) the results of the two groups in terms of total locomotor skills; (**e**) the results of the two groups in terms of total object control skills. Asterisks (**) indicates significant *p* (*p* < 0.01) and “0” the pre-test baseline.

**Figure 6 ijerph-19-01275-f006:**
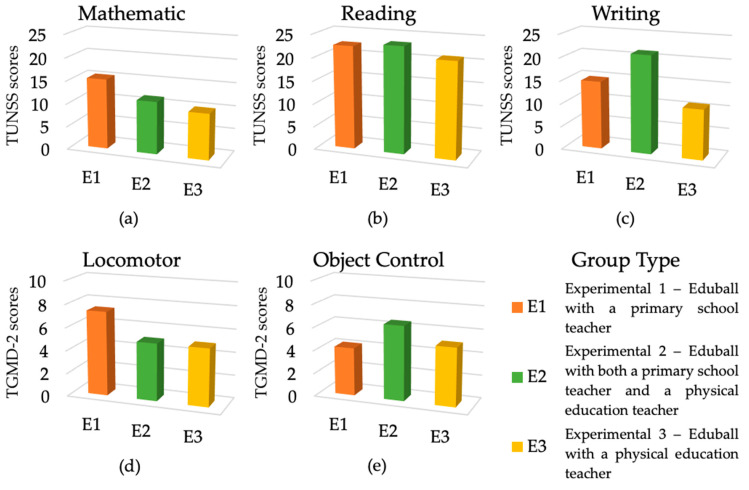
Results of the experiment in terms of cognitive and motor skills divided into three groups (Eduball 1 vs. Eduball 2 vs. Eduball 3) and expressed as difference scores. (**a**) The results of the two groups in terms of mathematical skills; (**b**) the results of the two groups in terms of reading skills; (**c**) the results of the two groups in terms of writing skills; (**d**) the results of the two groups in terms of total locomotor skills; (**e**) the results of the two groups in terms of total object control skills. Zero (“0”) indicates the pre-test baseline.

**Table 1 ijerph-19-01275-t001:** Mean and standard deviation of the control and experimental groups in the pre- and post-tests.

Skills	Control Group					Experimental Group 1				
	Pre-Test	Post-Test	*t*	*p*	*d*	Pre-Test	Post-Test	*t*	*p*	*d*
**Mathematic**	108.57 ± 12.16	119.86 ± 10.23	−4.71	<0.001	−1.03	110.33 ± 7.48	125.61 ± 6.4	−10.62	<0.001	−2.50
**Reading**	104.9 ± 12.89	119.71 ± 10.5	−5.01	<0.001	−1.09	103.61 ± 19.43	126.06 ± 8.94	−5.65	<0.001	−1.33
**Writing**	111.57 ± 9.45	119 ± 7.15	−3.80	<0.01	−0.83	110.17 ± 10.69	124.94 ± 9.17	−5.96	<0.001	−1.41
**Locomotor**	30.76 ± 5.89	34.81 ± 4.98	−2.84	0.01	−0.62	32.11 ± 5.94	39.44 ± 4.87	−4.98	<0.001	−1.17
*Run*	5.33 ± 1.98	6.52 ± 1.97	−1.88	0.08	−0.41	6.17 ± 1.58	6.83 ± 1.34	−1.41	0.18	−0.33
*Gallop*	5.29 ± 1.19	5.1 ± 1.95	0.41	0.69	0.09	5.94 ± 1.16	6 ± 1.97	−0.11	0.91	−0.03
*Hop*	6.05 ± 2.25	6.29 ± 1.76	−0.40	0.69	−0.09	5.44 ± 2.15	7.72 ± 1.81	−3.67	<0.01	−0.87
*Leap*	4.14 ± 1.35	4.67 ± 1.46	−1.21	0.24	−0.26	4.33 ± 1.37	4.67 ± 1.24	−0.79	0.44	−0.19
*Jump*	4.62 ± 2.01	6.1 ± 1.64	−2.75	0.01	−0.60	4.94 ± 2.04	6.78 ± 1.31	−4.51	<0.001	−1.06
*Slide*	5.33 ± 1.24	6.05 ± 1.88	−1.61	0.12	−0.35	5.28 ± 1.45	7.44 ± 0.78	−5.44	<0.001	−1.28
**Object Control**	27.1 ± 5.55	31.71 ± 6.42	−3.21	<0.01	−0.70	30.67 ± 6.26	34.83 ± 6	−3.15	0.01	−0.74
*Strike*	5.05 ± 2.2	6.29 ± 2.74	−1.71	0.10	−0.37	6.67 ± 2.38	5.78 ± 2.6	1.53	0.15	0.36
*Dribble*	3.1 ± 2.14	4.43 ± 2.09	−2.51	0.02	−0.55	3.5 ± 2.41	5.56 ± 1.82	−3.12	0.01	−0.74
*Catch*	4.62 ± 1.36	4.76 ± 1.18	−0.53	0.60	−0.12	4.17 ± 1.58	5.5 ± 0.79	−3.69	<0.01	−0.87
*Kick*	5.71 ± 1.55	6.05 ± 1.66	−0.94	0.36	−0.21	6.44 ± 1.38	6.67 ± 1.97	−0.51	0.61	−0.12
*Throw*	3.86 ± 1.8	4.81 ± 1.75	−2.12	<0.05	−0.46	4.5 ± 2.07	4.83 ± 2.01	−0.74	0.47	−0.17
*Roll*	4.71 ± 1.85	5.05 ± 1.94	−0.67	0.51	−0.15	5.72 ± 1.6	5.89 ± 1.45	−0.44	0.67	−0.10
**Skills**	**Experimental Group 2**					**Experimental Group 3**				
	**Pre-Test**	**Post-Test**	** *t* **	** *p* **	** *d* **	**Pre-Test**	**Post-Test**	** *t* **	** *p* **	** *d* **
**Mathematic**	110.75 ± 5.97	122.19 ± 6.27	−8.34	<0.001	−2.08	115.73 ± 8.05	125.73 ± 5.65	−5.93	<0.001	−1.53
**Reading**	103.13 ± 13.59	126.31 ± 9.14	−6.38	<0.001	−1.60	107.13 ± 15.14	128.07 ± 7.31	−5.73	<0.001	−1.48
**Writing**	103.81 ± 11.28	125.13 ± 11.71	−5.85	<0.001	−1.46	116.73 ± 7.51	127.6 ± 7.06	−4.69	<0.001	−1.21
**Locomotor**	33.75 ± 5.95	38.75 ± 3.99	−3.29	0.01	−0.82	34.6 ± 5.99	39.6 ± 4.4	−3.94	<0.01	−1.02
*Run*	6.06 ± 1.48	7.06 ± 1.57	−1.67	0.12	−0.42	6.47 ± 1.81	7.27 ± 1.39	−1.36	0.20	−0.35
*Gallop*	5.38 ± 1.5	6 ± 1.37	−1.11	0.28	−0.28	5.6 ± 1.3	7.13 ± 1.25	−2.62	0.02	−0.68
*Hop*	6.63 ± 2	7.25 ± 1.81	−1.25	0.23	−0.31	7 ± 2.17	7.53 ± 1.96	−1.14	0.27	−0.30
*Leap*	4.81 ± 1.38	5.31 ± 0.79	−1.23	0.24	−0.31	4.33 ± 1.45	5.4 ± 0.91	−2.87	0.01	−0.74
*Jump*	4.94 ± 2.29	6.88 ± 1.59	−3.18	0.01	−0.80	5.8 ± 2.04	6.67 ± 0.98	−1.55	0.14	−0.40
*Slide*	5.81 ± 1.28	6.13 ± 1.09	−0.69	0.50	−0.17	5.4 ± 1.5	5.67 ± 1.29	−0.70	0.50	−0.18
**Object Control**	28.56 ± 7.75	35.06 ± 3.7	−3.91	<0.01	−0.98	30.93 ± 5.48	36 ± 5.88	−4.07	<0.01	−1.05
*Strike*	5.5 ± 2.22	6.81 ± 1.87	−1.86	0.08	−0.47	6 ± 1.6	7.2 ± 1.82	−1.99	0.07	−0.51
*Dribble*	3.69 ± 1.96	5.88 ± 1.59	−3.35	<0.01	−0.84	4 ± 2.75	5.87 ± 2.1	−4.00	<0.01	−1.03
*Catch*	4.56 ± 1.26	4.56 ± 1.31	0.00	1.00	0.00	5.2 ± 0.86	5.33 ± 0.9	−0.52	0.61	−0.13
*Kick*	5.06 ± 1.81	6.5 ± 1.71	−2.59	0.02	−0.65	5.93 ± 1.53	6.33 ± 1.68	−0.75	0.47	−0.19
*Throw*	5 ± 2.25	5.38 ± 1.93	−0.53	0.61	−0.13	4.87 ± 2.26	5.47 ± 2.17	−1.29	0.22	−0.33
*Roll*	4.75 ± 1.61	5.5 ± 1.63	−1.34	0.20	−0.34	4.93 ± 1.39	5.4 ± 1.59	−1.10	0.29	−0.28

**Table 2 ijerph-19-01275-t002:** Analysis of covariance (ANCOVA) for the cognitive and motor skills by group condition (no-Eduball vs. Eduball group). Result of pre-test was set as the covariate.

Skills	*SS*	*SM*	*F*	*p*	*η_p_* ^2^	*OP*
**Mathematic**	129.91	129.91	3.18	0.08	0.05	0.42
**Reading**	745.75	745.75	10.33	<0.01	0.13	0.89
**Writing**	764.97	764.97	10.98	<0.01	0.14	0.90
**Locomotor**	195.10	195.10	10.62	<0.01	0.14	0.89
*Run*	4.45	4.45	1.73	0.19	0.03	0.25
*Gallop*	23.80	23.80	7.81	<0.01	0.10	0.79
*Hop*	19.95	19.95	6.48	0.01	0.09	0.71
*Leap*	2.47	2.47	1.75	0.19	0.03	0.26
*Jump*	4.99	4.99	2.64	0.11	0.04	0.36
*Slide*	2.42	2.42	1.08	0.30	0.02	0.18
**Object Control**	70.01	70.01	2.91	0.09	0.04	0.39
*Strike*	0.05	0.05	0.01	0.92	0.00	0.05
*Dribble*	19.24	19.24	5.96	0.02	0.08	0.67
*Catch*	2.15	2.15	1.87	0.18	0.03	0.27
*Kick*	2.58	2.58	0.95	0.33	0.01	0.16
*Throw*	0.01	0.01	0.00	0.95	0.00	0.05
*Roll*	2.70	2.70	1.05	0.31	0.02	0.17

**Table 3 ijerph-19-01275-t003:** Analysis of covariance (ANCOVA) for the cognitive and motor skills by group condition (Experimental 1 vs. Experimental 2 vs. Experimental 3). The pre-test was set as the covariate.

Skills	*SS*	*SM*	*F*	*p*	*η_p_^2^*	*OP*
**Mathematic**	119.40	59.70	2.42	0.10	0.10	0.46
**Reading**	14.41	7.20	0.11	0.89	0.01	0.07
**Writing**	31.13	15.57	0.18	0.83	0.01	0.08
**Locomotor**	11.86	5.93	0.35	0.71	0.02	0.10
*Run*	1.63	0.81	0.39	0.68	0.02	0.11
*Gallop*	13.13	6.56	2.62	0.08	0.10	0.49
*Hop*	6.85	3.42	1.14	0.33	0.05	0.24
*Leap*	5.20	2.60	2.51	0.09	0.10	0.48
*Jump*	1.17	0.58	0.37	0.70	0.02	0.11
*Slide*	29.16	14.58	12.87	<0.001	0.36	1.00
**Object Control**	14.00	7.00	0.35	0.71	0.02	0.10
*Strike*	24.81	12.40	2.84	0.07	0.11	0.53
*Dribble*	0.65	0.32	0.11	0.90	0.00	0.07
*Catch*	8.64	4.32	4.25	0.02	0.16	0.71
*Kick*	1.47	0.73	0.24	0.79	0.01	0.09
*Throw*	2.04	1.02	0.30	0.74	0.01	0.09
*Roll*	0.53	0.27	0.12	0.89	0.01	0.07

## Data Availability

The data presented in this study are openly available in the Open Science Framework at https://osf.io/fhqpj (accessed on 20 December 2021).

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
