# Peer review of "Everyone Can Implement Eduball in Physical Education to Develop Cognitive and Motor Skills in Primary School Students"

_ijerph, 2022, doi:10.3390/ijerph19031275_

Round 1
Reviewer 1 Report
There is one spelling mistake in the following sentence: "First we tasted fundamental motor skills.........." on page 151. tasted - replace by tested.
This article attracted my attention and solves quite an interesting issue concerning who can/may teach lessons with Eduball. Congratulation! However, I miss a little more physiology approach to the introductory part of the article.
I suggest to change the title of the article in the following way:
- Since the name of the subject at school is „Physical Education (or possibly Physical Education and Sport), I recommend to insert the word EDUCATION into the title of the article:
„Everyone can: Physical EDUCATION Classes with Eduball...................“
I also recommend the authors to give the reader fundamental information on the physiology of cognitive processes:
The reader should understand the neural basis underlying conscious perception. Using the auditory and visual sensory modalities as models, we investigate where sensory objects are encoded in the brain, how they are represented by neuronal activity; and how this representation is shaped by learning. This information is necessary also for the understanding of the role of a teacher (or a coach) in learning processes. The brain is the most complex organ of our body. It enables us to do complex behavioral tasks and find ready-to-go solutions for abstract problems. The brain uses many different subsystems to respond in an adequate manner to a stimulus from the environment. One of these systems is the cholinergic network located in the basal forebrain and extending from there through the entire cortex. Learning and cognition heavily depend on an intact cholinergic system, which guarantees attention to important and relevant cues from the outside world and also insures subsequent memory formation of these cues to create an integrated and reliable experience for adequate behavior in later life.
Author Response
Dear Reviewer,
Please see the attachment.
Sincerely,
Michal Klichowski
and Sara Wawrzyniak

Reviewer 2 Report
Thank you for submitting your paper to this journal. I very much enjoyed reading it and think that it has the potential to be of interest to readers from a wide range of disciplines (i.e., physical education, primary education, motor learning etc.). However, that said, after reading the manuscript it is clear that further work is need on the written communication; specifically, the English language and style. Also, greater clarity is required in aspects of the methodology. I hope you find the comments useful for your resubmission.
Comments:
The manuscript requires a review of written communication and a full proofread (see below) – the following comments are examples of errors and inconsistencies and are not exhaustive.
- The manuscript contains general ‘phrasing’ issues in a great many sentences, e.g., page 1, line 4 (“…Students no Matter which Teacher Conduct them”) and page 5, line 181 (“This commonly use in Poland test…”).
- Terminology and acronyms are not consistent in the manuscript, e.g., CT and PET are referred to on page 2 (lines 46 and 50) but are not used on page 14 (line 291; primary school teacher and physical education teacher).
- Aspects of the paper includes poor spelling (page 4, line 151; tasted/tested) and inconsistent language (page 4, line 170; child/student).
- Sentences that start directly with a conjunction should be avoided (e.g., page 2, line 49; But).
- Small sentences which do not stand-alone as a sentence should be avoided (e.g., page 2, line 49; “But there is also other possibility”).
- Avoid abbreviations (e.g., page 14, line 327; math/mathematics).
- The clarity of the dates in the manuscript could be made clearer (i.e., page 4, line 130; the phrase “week 4-10.10.15” could be misconstrued as ‘week 4 to 10’ in the tenth month of 2015.
- Tables and figures are generally presented in an acceptable manner, however, labels to indicate what the numbers represent would be useful for the Y axis on most figures.
- The reference list needs to be proofread in terms of accuracy and punctuation (page 17, line 455; J Sci Med Sport requires punctuation).
- Methodology – Aspects of the ‘procedure’ section of the methodology could be made explicit. For example: What kind of teaching took place in the control group? (Was this command-orientated or discovery-/experiential learning-based?). How did the collaboration sessions work? (Was the teaching split equally or did the CT or PET take the lead?). Greater description of methodological aspects such as these are required so they can be replicated in future studies.
Author Response

(The authors gave the same response as above.)

Reviewer 3 Report
I find it interesting and innovative work. Even the graphics have new features.Perhaps the activity they propose is not well known or practiced, but from this
forum it can be disseminated and motivated to practice it. I recommend that you follow this line of research.
Author Response

(The authors gave the same response as above.)
